# Inducing Stronger Object Representations in Deep Visual Trackers

## Abstract

Fully convolutional deep correlation networks are integral components of state-of-the-art approaches to single object visual tracking. It is commonly assumed that these networks perform tracking by detection by matching features of the object instance with features of the entire frame. Strong architectural priors and conditioning on the object representation is thought to encourage this tracking strategy. Despite these strong priors, we show that deep trackers often default to "tracking by saliency" detection – without relying on the object instance representation. Our analysis shows that despite being a useful prior, salience detection can prevent the emergence of more robust tracking strategies in deep networks. This leads us to introduce an auxiliary detection task that encourages more discriminative object representations that improve tracking performance.

## 1 Introduction

Single object visual tracking is a well studied, classical problem in computer vision. It is defined as follows: given a video sequence and an initial annotation (commonly an axis aligned bounding box) that localizes some target object of interest, produce the object annotations corresponding to the same object for the remaining frames of the sequence. Because tracking is such a generic visual capability, it is of interest in both theoretical and practical computer vision. Object tracking is found in a wide range of computer vision applications including autonomous vehicles, surveillance, and robotics. In theoretical computer vision, object tracking has been used to develop several theories of invariant object representations (Lim et al., 2005). Tracking is also a fundamental capability found in many natural visual systems. Prior to the availability of large annotated video datasets most video-based feature learning models were trained using unsupervised learning objectives, many of them inspired by Slow Feature Analysis (SFA) (Wiskott & Sejnowski, 2002). With the availability of large annotated video datasets such as ILSVRC 2015 Object Detection from Video Challenge (Russakovsky et al., 2015) and YouTube-Bounding Boxes (Real et al., 2017), it has become popular to learn features directly by optimizing tracking objectives. Deep learning approaches have been successfully applied to attain state-of-the-art results on several tracking benchmarks such as VOT2018 (Kristan et al., 2017).

Though seemingly similar to detection, in that trackers receive images as input and are expected to output bounding boxes, neural trackers are tasked to solve a subtly different problem from detectors. Some key differences include:

- Siamese trackers are conditioned on the target image, most detectors are not. One notable exception is the recent work on "target driven instance detection" that uses a Siamese architecture similar to those used in tracking to detect particular *instances* of objects (Ammirato et al., 2018). Whereas detectors represent all classes of interest in theirs weights, trackers are expected to form a representation of the target object using a template image.
- Siamese trackers often incorporate a small displacement prior by restricting the search area to be near the previous bounding box.

The latter constraint often results in center-cropped images of the target input to the "search" branch of the Siamese network. Furthermore, the action of cropping restricts the presence of background objects in the search image. This means that the network can often achieve a low tracking loss by detecting the *most salient object in the center of the search image.*

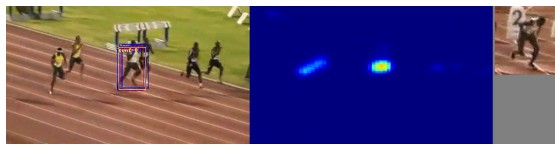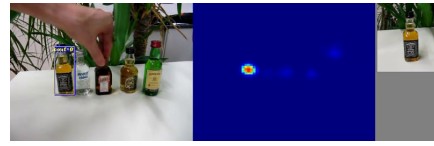

Figure 1: The output of our conditional detector. The two examples show: Left - the detector input. Middle - the heatmap output of our detector depicting the likelihood of the target object centered on each pixel. Right - the "target" template image specifying the object instance to detect.

Currently the prevalent approach adopted by many neural trackers is that of "tracking by detection" (Breitenstein et al., 2009).Recently, deep correlation Siamese networks have been central components in state-of-the-art tracking approaches (Li et al., 2019; 2018; Bertinetto et al., 2016). We refer to these object tracking networks as "Siamese trackers". These networks functionally resemble object detection networks (such as Fast-RCNN (Ren et al., 2015) and YOLO (Redmon et al., 2016)) in that they localize semantic objects by producing bounding box outputs. Unlike object detection networks, which are trained to detect object categories, Siamese trackers are trained to detect the object using the provided template in a video sequence. These networks are trained end-to-end on video datasets on a per-frame detection loss. The network architectures used by Siamese trackers emphasize tracking by detection via correlation of the target and search images in a learned feature space (Bertinetto et al., 2016; Held et al., 2016). However, because the training objective is minimized end-to-end, networks need not necessarily optimize the tracking objective by feature matching. The aim of this paper is to study these trackers' reliance on the target representation and show that Siamese trackers often default to tracking by center saliency detection (Mahadevan & Vasconcelos, 2012) – without necessarily forming a discriminative target representation. In other words, a simple strategy that neural trackers can use to get good performance is to track the object that is most *salient* in the search region instead of attempting to rely on matching features with the target template (see Figure 4). On the basis of these insights, we introduce an auxiliary instance driven detection objective Ammirato et al. (2018) that improves tracking performance over our baseline. An example of our detection output is shown in Figure 1.

## 2 PRIOR WORK

Like many fields in computer vision, single object tracking has become dominated by deep learning approaches. However, unlike image classification (Krizhevsky et al., 2012) and object detection (Sermanet et al., 2013), which were arguably the first successes of deep learning on large scale natural image datasets, object tracking had been relatively untouched by deep learning until 2016 with the publications of GOTURN (Held et al., 2016) and MDNet (Nam & Han, 2016). These seminal works have spawned successors, many of which still dominate tracking performance in the most popular benchmarks (Kristan et al., 2017; Wu et al., 2013; Valmadre et al., 2018). This shift was perhaps triggered by the release of ImageNet Object Detection from Video in 2015 (Russakovsky et al., 2015), the first per-frame annotated large scale video dataset. GOTURN is an example of the so called "tracking by detection" approach, where tracking is performed by recursive detection on video frames. GOTURN and it's derivatives implement Siamese-networks to directly regress the bounding box coordinates of the target object. These networks typically consists of four main elements: the target network, the search network, the join layer, and the output network. The target and search networks compute feature maps of the corresponding to the target object and the search region, respectively. These feature maps are combined in the *join layer*; most recent trackers use convolution/correlation to combine the feature maps (Bertinetto et al., 2016). Finally, the combined feature maps are fed to the output layer which consists of a trainable network that outputs the final bounding box or bounding box proposals (Li et al., 2018; 2019; Held et al., 2016). Most recently SiamRPN (Li et al., 2018), SiamRPN++ (Li et al., 2019), and their derivatives have been the dominant state-of-the-art approaches to single object tracking. These works combine region proposals popular in the detection literature with Siamese tracking. Later we will introduce detection as an auxiliary task for tracking. Training a model to jointly track and detect has recently been proposed in (Feichtenhofer et al., 2017). However, this work is quite orthogonal to ours as it does not consider object *instance* representation and tracks by temporally linking detected instances.

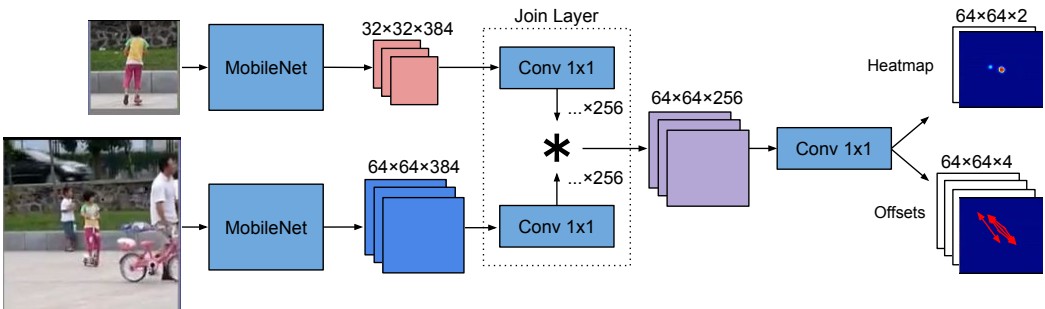

Figure 2: The core Siamese tracker takes the target and search images as input and outputs the bounding box proposal heatmap and offsets.

# 3 BASIC TRACKING MODEL

Our basic tracking model takes inspiration from recent works starting with GOTURN (Held et al., 2016) and culminating in SiamRPN++ (Li et al., 2019). Our model uses a lightweight backbone network (MobileNetV2), and is somewhat simpler than recent state-of-the-art models. Nevertheless, our model contains key elements that are common to most recent deep trackers and therefore we believe our conclusions also apply to these models. Although the model doesn't outperform state-of-the-art, it attains competitive performance. This section details the architecture of the tracking network, the targets and loss it is trained with, the training data sampling method, data augmentation, and post-processing of the model outputs.

## 3.1 TRACKING NETWORK

The overall architecture of our tracking network is shown in Figure 2. The tracker takes two images as input: the *target* image ($x_t$) is the centered crop of the object instance to be tracked and the *search* image ($x_s$) is a crop of the image in which the target is to be localized. The frame corresponding to the target image precedes the frame corresponding to the search image. After re-sizing, the target and search image have shapes $127 \times 127 \times 3$ and $255 \times 255 \times 3$, respectively. We use the same aspect preserving cropping mechanism detailed in (Bertinetto et al., 2016). Feature maps corresponding to both images are computed using a pre-trained 0.75-MobileNet-224 backbone network (Howard et al., 2017). Because the weights are shared between the target and search branches the two images are encoded using the same function (i.e. the backbone network) denoted by $f_\theta$. Feature maps corresponding to the target and search images have shapes $32 \times 32 \times 384$ and $64 \times 64 \times 384$, respectively. We denote these feature maps as $z_t^i = f_\theta(x_t)$ and $z_s^i = f_\theta(x_s)$, where $i$ is the feature plane index. Next we project these feature maps using a trainable $1 \times 1$ convolution to 256 dimensions. The resulting target and search features are combined by convolving each search feature map with the corresponding target feature map with zero-padding ("*same*" convolution), to obtain a combined activation map $z^i = z_t^i * z_s^i$ that has shape $64 \times 64 \times 256$. We refer to this operation as *cross-convolution*. A similar operator for combining the target and search features was first introduced in (Bertinetto et al., 2016) and has since been employed in many recent tracking architectures. Finally the resulting feature map is projected to six channels using a trainable $1 \times 1$ convolution layer, resulting in a final output of size $64 \times 64 \times 6$. It is worth noting, we have found that inserting additional trainable layers after the cross-convolution stage significantly reduces performance.

The tracker outputs $n$-bounding box proposals and their respective confidence (where $n$ is a hyper-parameter we set to 5). Our bounding box proposal parameterization is inspired by the fully convolutional heatmap and offset parameterization introduced for points in (Papandreou et al., 2017), which we adapt to bounding boxes. The first two of the six output channels represent the center location of the bounding box. These two channels are used to represent the positive and negative classes of the binary heatmap. These channels are trained with a binary cross-entropy loss with targets that are constructed by placing a fixed radius (10 pixels) circle of ones centered at the location of the ground truth bounding box, the heatmap targets are set to zero elsewhere. The remaining

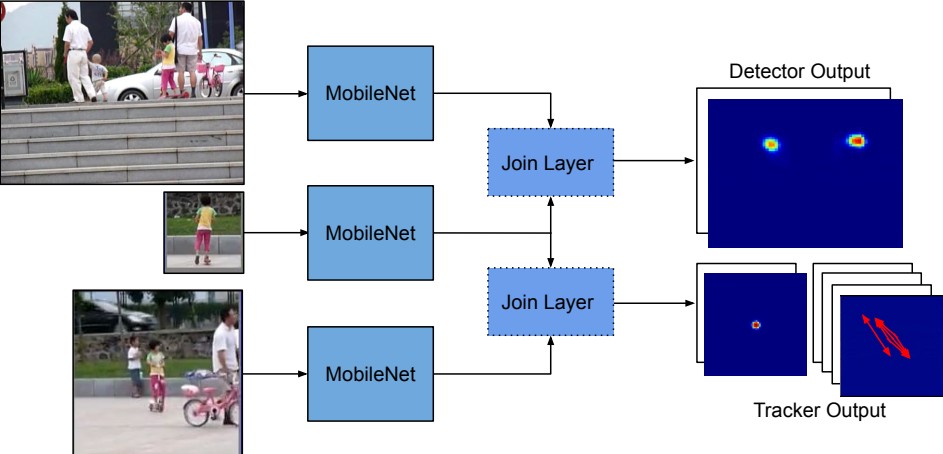

Figure 3: Joint tracking and detection model.

four feature maps are used to represent the spatial offsets to the top-left and bottom-right bounding box corners relative to their location in the feature map. The offset channels are trained with an $L_1$ regression loss. To produce the final bounding box output we enumerate the top $n$ modes in the heatmap by using non-maximal suppression with a fixed square window ($6 \times 6$ pixels) centered on each mode. The mean offset is computed within each window to give the offset corresponding to each bounding box corner. In contrast with (Papandreou et al., 2017), we do not use Hough voting to aggregate the offsets.

The model is trained on short sub-sequences taken from the ILSVCR 2015 Detection from Video (Russakovsky et al., 2015) and the GOT-10k (Huang et al., 2018) video datasets. To encourage the network to learn invariant representations of the target across long time scales, the target and search images are sampled such that they are unlikely to be temporally proximal frames. Suppose a given dataset sequence consists of $T$ frames and we sample a sub-sequence of length $\tau$ starting at index $t$ such that $t + \tau < T$. The target image is selected uniformly from the interval $[0, t - 1]$, to respect the fact that the target template must appear *before* the search image. Networks were trained with several target image selection mechanisms including always using the first image from the sequence and sampling random causal targets. The latter lead to better performance, likely due to increased diversity. During evaluation, the target image is cropped from the first frame using the ground truth bounding box and held it fixed for the rest of the sequence. We apply the following data augmentation to both target and search images during training: random translation and scaling, as well as random perturbations to the contrast, hue, and saturation.

The model was trained using the Adam optimizer (Kingma & Ba, 2014) for $1e7$ steps with an initial learning rate of $10^{-4}$ which was reduced by a factor of 10 at $9.5e6$ steps. The heatmap and regression loss are combined using a weighted sum. The weight corresponding to the heatmap loss is set to 1.0 and regression weight is set to 0.3.

### 3.2 PROPOSAL SELECTION

Proposal selection is performed using the same heuristics as those presented in (Li et al., 2018). Namely, a penalty for abrupt changes between adjacent frames in bounding box aspect ratio and center location is imposed. The penalty is applied to the scores output by the heatmap channel of the network before outputting the final bounding box prediction corresponding to the maximal score of the penalized heatmap.

## 4 INDUCING DISCRIMINATIVE TARGET REPRESENTATIONS

Siamese trackers may learn a variety of strategies that minimize the per-frame tracking loss. We will show in Section 4 that center *saliency* detection in the neighborhood of the previously tracked

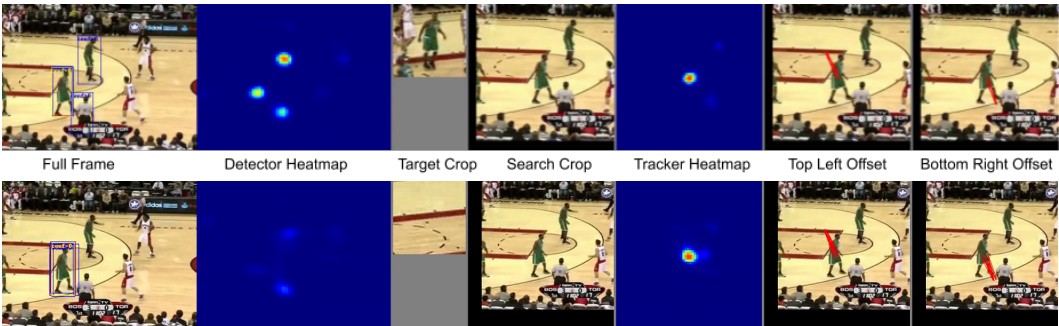

Figure 4: Joint tracking and detection model output. Top: ground truth target crop if input to the target branch. Bottom: a random patch from the first frame is input to the target branch.

bounding box attains good performance if the sequence consists of a smoothly moving un-occluded object in an uncluttered background (Mahadevan & Vasconcelos, 2012). In fact, such a tracking strategy is more robust to changes in appearance of the object than an *appearance matching* strategy. The center saliency strategy does not require a discriminative representation of the target object instance, leaving the target feature branch of the network virtually unused. Unfortunately, the vast majority of sub-sequences sampled during training contain smoothly moving objects with little or no occlusion. Several works (Li et al., 2019; Held et al., 2016) apply data augmentation to the search crop, which reduces center bias learned by the network. Nevertheless manual data augmentation alone limits the types of appearance variations that the tracker can learn. Furthermore, despite discouraging discriminative target representations, centering the target in the search window is an important prior that ultimately leads to better tracking performance.

As mentioned in Li et al. (2019), random translation augmentations during training are crucial to prevent the network from overly relying on simple priors, such as center bias and saliency detection, while at the same time improving generalization of search and target representations. Neverthe-less, priors that rely on temporal coherence are indeed useful, and so we seek a mechanism that encourages discriminative target representations, while allowing the tracker to utilize temporal co-herence. Our proposed solution is to add *instance driven detection* as an auxiliary task. The joint tracker/detector network can be implemented by adding another branch and join layer to the tracking network as shown in Figure 3. The detector branch is similar to the search branch *except for the type of data it is trained on*. The detector is trained on temporally decoherent, un-cropped frames. Unlike the tracker, it cannot rely on center saliency detection, and must rely on a more discriminative target representation to minimize the detection loss. Because the target representation is shared between the tracker and detector, increasing the weight on the detection task encourages a more discrimina-tive representation of the target (we use a weight of 1.0 in our experiments). Such a representation is crucial for avoiding failures in rare but critical portions of the video where the tracker cannot rely on center saliency alone (e.g. occlusion, presence of other salient objects, etc). The detector only outputs a heatmap which is trained to respond at the center of the target object's bounding box in the search image. We omit the offset channels and regression loss in the detector branch.

Figure 4 depicts a representative output of the jointly trained model (tracker-detector) for a frame of the *basketball* sequence in VOT2018. The Full Frame is the original frame from the dataset. Proposed bounding boxes, output by the tracker are overlaid on the Full Frame image in blue. The final bounding box output is depicted in red. After re-sizing to standard resolution of $255 \times 255$ the Full Frame is input to the detector, which outputs the Detector Heatmap. In order to visualize two-channel heatmaps, the soft-max is computed across the two channels. The Target Crop shows the image that is input to the target branch and held fixed for the entire sequence. The remaining three images visualize the output of the tracker. The Detector Heatmap, defined over the full frame, responds strongly to the non-target players. However, the Tracker Heatmap suppresses it's response to the non-centered players. Because the detector sees only full frames, it must rely on appearance alone to detect the target, thereby encouraging a more discriminative feature representation of the target. To further confirm this effect we applied the same model to the *basketball* sequence again, however now we input a randomly cropped patch (not containing the image of the target player)

to the target branch. The Search Crop was still initialized using the ground truth bounding box corresponding to the first frame, ensuring that the target is at the center of the search window. Despite having input the incorrect Target Crop, the tracker still outputs large logit scores at locations corresponding to the tracked player, located at the center of the Search Crop. Also note that the final bounding box output by the tracker is approximately the same (however the proposals are less diverse). In contrast, the logit scores of the detector are much smaller at the player locations, implying that the detector is more sensitive to the target representation. To confirm that the tracker is capable of tracking by saliency detection alone, we evaluated the tracker while feeding random patches extracted from the first frame into the target branch. Although the performance significantly decreased (last row in the Table of Figure 5), the tracker still tracks the target in the vast majority of frames and even maintains no failures in some sequences.

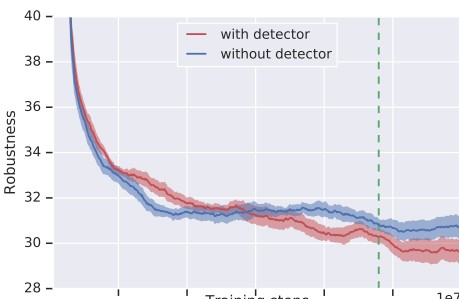

| Model | Robustness | Accuracy |
|---|---|---|
| SiamRPN++ | 14.87 | 0.5915 |
| SiamMask | 15.40 | 0.5889 |
| SiamRPN | 18.39 | 0.5639 |
| Ours | 17.68 | 0.5562 |
| Ours (no detector) | 26.61 | 0.5361 |
| Ours (random target) | 38.50 | 0.4740 |

Figure 5: *Left*: Mean and standard error plot of VOT2018 robustness versus training iteration. The error bars were obtained using six random seeds. The dashed line represents the training iteration when the learning rate was reduced. *Right*: Table comparing final VOT2018 performance to recent state-of-the-art Siamese trackers.

The effect on the number of tracking failures of training with the detection auxiliary task is shown in the plot in Figure 5. The y-axis depicts the supervised robustness of VOT2018. The robustness is defined as the mean number of failures weighted by sequence length in "supervised evaluation" on the entire dataset (60 videos). A failure is triggered when the output bounding box has zero IoU with the ground truth. The tracker is reset after a failure by the evaluation toolkit (Kristan et al., 2017; Huang et al., 2018). The training curves were obtained by averaging the smoothed training curves corresponding to six random initializations for models trained with and without the detector. The standard error curves are shown in light colors. Each point on the curve corresponds to a *full supervised evaluation* on the VOT2018 dataset. This corresponds to a total of 563 full VOT2018 evaluations for each random seed (3378 total evaluations) as opposed to a single evaluation typically reported in the tracking literature. In the early stages of training, the detection sub-task actually hinders performance, however it helps reduce the number of failures later in training. Our hypothesis is that in the early phase of training, the tracker learns to track by saliency, the detection task actually interferes with learning this strategy. In the later phase of training, the tracker begins to rely more heavily on appearance matching in features space. This is when a discriminative target representation actually proves to be useful. In the following section we performing additional analysis to support these conclusions. For completeness, the table in Figure 5 compares our best model with an official implementation of SiamRPN++ [1] on the VOT2018 dataset.

## 5 PERTURBATION ANALYSIS

The improved robustness achieved by adding the detector can be attributed to several factors. Detection may induce a more robust target representation, or it may force the tracker to rely less heavily on saliency detection, or both. In order to measure these potential effects, we evaluate the sensitivity of our architecture to variations in the input conditioning of our network; namely the target location, search location, and target time step. The first experiment is shown in Figure 6, whereby we demonstrate the sensitivity of our tracker to perturbations in the search window location using

---

[1]https://github.com/STVIR/pysot

the following procedure. First, we sample a pair of adjacent frames from the VOT dataset and we crop a target representation on frame $t$ using the ground-truth target location. Next, we crop a search window on the next frame $t+1$ using the target location from frame $t$, and we infer the predicted box location and calculate IoU to the ground truth box at frame $t+1$. We then repeatedly crop additional search windows with displacements in the horizontal and vertical directions, where displacements are scaled by a box size factor (box width$_t$ + box height$_t$) $/2$. We then calculate a heatmap of the IoU change as a function of this displacement, normalized by IoU at the center displacement $(0,0)$. We then repeat this procedure for all adjacent frame pairs for all sequences in VOT and average the heatmap responses. The resultant heatmap is rendered in Figure 6.

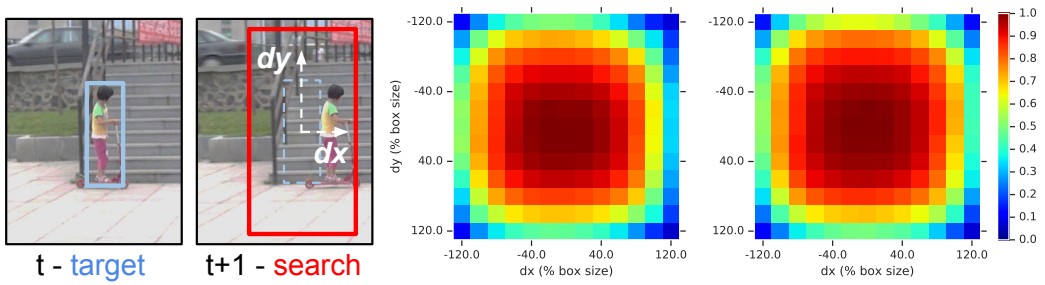

Figure 6: Search window location sensitivity analysis of our network. Left: sampling procedure, Middle: fraction of IoU (normalized by the IoU at displacement $(0,0)$) for our network without detector, Right: fraction of IoU for our network with detector.

As shown in Figure 6, our network is able to infer the next-frame bounding box location despite large displacements of the search window. This is largely a result of mitigation of center bias via translation augmentation of the search window during training. Note that a 100% shift in bounding box location (relative to the bounding box size) is an extremely large spatial displacement at 24fps. At search window displacements larger than approximately 150%, the target is no longer visible in the search window, and so as expected the IoU drops to zero. Additionally, there is no significant difference between these response maps of our architecture trained with and without the detector. This suggests that the improved performance obtained by adding the detector cannot be attributed to reduced center bias in the tracker.

We also analyze the sensitivity of the network to target crop location. The procedure is similar to the search window experiment above, except we instead keep the search window constant and vary the position of the target crop. The result of this experiment is shown in Figure 7 and demonstrates interesting tracking behavior. First, as the target moves by small displacements the tracker shifts it's output bounding box by the same fraction, causing an expected sharp roll-off in IoU (since we calculate IoU to unmodified target location). This indicates that the tracker is indeed paying attention to the target crop in order to infer the object extent. However, if the target crop is moved far enough in either direction (approximately 50% of the box size), the network begins to ignore the target and

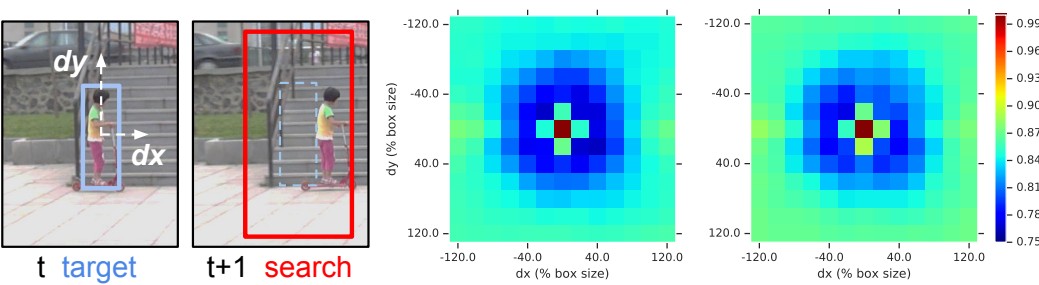

Figure 7: Target crop location sensitivity analysis of our network. Left: sampling procedure, Middle: fraction of IoU for our network without detector, Right: fraction of IoU for our network with detector.

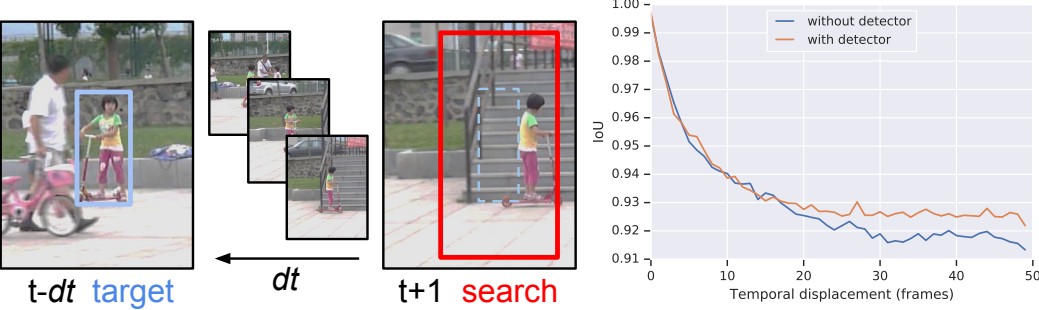

Figure 8: Target crop temporal sensitivity analysis of our network. Left: sampling procedure, Right: mean fraction of IoU for our network without and without detector (normalized by the iou at $dt = 0$.)

switches to *saliency detection*. In this case, the IoU actually increases from the central low to some background constant.

During frame transitions with poor target quality, saliency detection is often a desired characteristic as it allows the network to continue tracking until the target resembles it's template. However, a drop of approximately 15% (difference in background intensity in Figure 7) of baseline IoU highlights the accuracy lost when relying on saliency alone; since the network does not have a precise representation of the tracked target, it tends to include other salient features from the background. Note that our jointly trained tracker/detector model outperforms the baseline tracker, where the background IoU lost due to saliency is slightly less.

Lastly, we examine our tracker's IoU sensitivity to the temporal displacement between the target frame and the search frame. Similar to the target displacement experiment above, we use a constant search window and instead choose a ground truth target bounding box some $dt$ frames in the past. The further back in time the target is cropped, the more "stale" it is. We expect that as the temporal displacement between the current frame and the template grows, the visual appearance of the template becomes a poorer representation of the specific object to track, and we correspondingly expect some drop in IoU as a result. Here, we normalize the IoU by the IoU measured when a target at $dt = 1$.

The effect of target staleness on average IoU is shown is shown in Figure 8. Firstly, both of our network variants show an asymptotic drop in performance. In the large displacement regime, when direct template matching is impossible, the network must use a more abstract representation of the target object. Bounding box accuracy suffers in this regime as the network can no longer use the template to infer the true object extent. Instead, it must rely on salient features in the search image. However, some of this performance decrease is mitigated by our network variant that includes the detection branch.

## 6 CONCLUSION

In this work we presented a thorough study of the representations learned by Siamese trackers. We showed that these trackers often rely on saliency detection despite being designed to track by template matching in feature space. We proposed an auxiliary detection task that induces stronger target representations and improves performance. These findings were empirically validated by performing perturbation experiments.

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
