# OpenReview forum: "Inducing Stronger Object Representations in Deep Visual Trackers"
_ICLR.cc/2020/Conference — Reject_

### Official Review · AnonReviewer2 · 2019-10-19
**Official Blind Review #2**

**Rating:** 3

**Review:**

This paper investigates representations learned by Siamese trackers. The paper argues that existing trackers rely on saliency detection despite being designed to track by template matching in feature space. An auxiliary detection task is proposed to induce stronger target representations in order to improve tracking performance. Experiments are performed on VOT2018 tracking dataset.

The paper investigates an interesting and active research problem of stronger object representations for deep visual object tracking. However, the proposed solution of just integrating an additional detection task branch within the Siamese tracking architecture is naive. The main idea of integrating instance driven detection as an auxiliary task is borrowed from [1].  [1] also utilizes a Siamese architecture that is similar to the ones generally used in visual object tracking to localize particular instances of objects. Therefore, the novelty of the proposed tracking approach is limited.

Some recent works, such as [2, 3] have also investigated a similar problem of richer object representations for deep visual tracking. These approaches are desired to be discussed and empirically compared in order to fully validate the strength of the proposed approach.

The paper shows some qualitative analysis. However, most of it is limited to just few frames of an image sequence. Tracking datasets, such as VOT and OTB, provide additional analysis tools (i.e., attribute analysis) to thoroughly evaluate visual trackers. Such analysis is missing in the paper. For instance, the main argument of this paper is that current approaches rely on center saliency and likely struggle in the presence of occlusion. How does the proposed approach fare, compared to SOTA, on the subset of VOT image sequences that are labeled with occlusion?

On page 3, it is stated that "Our model uses a lightweight backbone network (MobileNetV2), and is somewhat simpler than recent state-of-the-art models  .....................  Although the model doesn’t outperform state-of-the-art, it attains competitive performance." The reviewer does not fully agree with this statement. A comprehensive empirical evaluation is crucial to fully access the merits of the contributions. State-of-the-art visual object trackers [2, 3, 4] achieve competitive tracking performance while being computationally efficient and fast. Therefore, a proper state-of-the-art comparison is desired to compare the proposed tracker with SOTA methods. Further, currently experiments are only performed on the VOT2018 dataset. The reviewer recommends to perform additional experiments on other large-scale datasets, such as TrackingNet [5] and Lasot [6] and compare the performance with SOTA methods that are also investigating the problem of richer object representations for tracking.

[1] Phil Ammirato, Cheng-Yang Fu, Mykhailo Shvets, Jana Kosecka, Alexander C. Berg: Target Driven Instance Detection. CoRR abs/1803.04610 (2018).
[2] Martin Danelljan, Goutam Bhat, Fahad Shahbaz Khan, Michael Felsberg: ATOM: Accurate Tracking by Overlap Maximization. CVPR 2019.
[3] Goutam Bhat, Martin Danelljan, Luc Van Gool, Radu Timofte: Learning Discriminative Model Prediction for Tracking. CoRR abs/1904.07220 (2019).
[4] Bo Li, Wei Wu, Qiang Wang, Fangyi Zhang, Junliang Xing, Junjie Yan: SiamRPN++: Evolution of Siamese Visual Tracking With Very Deep Networks. CVPR 2019.
[5] Matthias Müller, Adel Bibi, Silvio Giancola, Salman Al-Subaihi, Bernard Ghanem: TrackingNet: A Large-Scale Dataset and Benchmark for Object Tracking in the Wild. ECCV  2018.
[6] Heng Fan, Liting Lin, Fan Yang, Peng Chu, Ge Deng, Sijia Yu, Hexin Bai, Yong Xu, Chunyuan Liao, Haibin Ling:
LaSOT: A High-Quality Benchmark for Large-Scale Single Object Tracking. CVPR 2019.



**Experience Assessment:**

I have published in this field for several years.

**Review Assessment: Checking Correctness Of Derivations And Theory:**

I assessed the sensibility of the derivations and theory.

**Review Assessment: Checking Correctness Of Experiments:**

I carefully checked the experiments.

**Review Assessment: Thoroughness In Paper Reading:**

I read the paper thoroughly.

---

### Official Review · AnonReviewer1 · 2019-10-22
**Official Blind Review #1**

**Rating:** 3

**Review:**

The paper examines the performance of Siamese single-object trackers. The authors claim that state-of-the-art Siamese trackers mostly rely on saliency detection in the center of the search window while ignoring the target instance representation and propose an additional object detection branch during training to mitigate this effect.

Strengths:
+ Analysis how perturbation target and search image influence tracking performance
+ Demonstrates that including detection objective during training improves performance

Weaknesses:
- Makes claims about Siamese trackers in general but only experiments with authors’ architecture
- Analyses in Figs. 6–8 not very helpful/conclusive
- No test case where capturing appearance is important for tracking and authors’ approach helps
- No comparison of tracker trained without detection objective on real vs. random targets

The paper is well motivated and has a clear hypothesis. However, its execution unfortunately leaves a lot of room for improvement and getting it into acceptable shape would require a major revision. Some details on my main criticisms:

Claim is way too general. The authors state that Siamese trackers in general suffer from the center bias problem. However, the authors do not analyze any other trackers than their own one. I understand that training a state-of-the-art tracker with the additional object detection branch could possibly require resources beyond what’s available to the authors. However, it’s not clear to me why the authors do not use the pre-trained version of at least one or two existing trackers to demonstrate their shortcomings (like Fig. 4)

Analyses in Fig. 6–8 are not very helpful. The only real effect the detector shows in Figs. 6–8 is a tiny improvement in IoU in Fig. 8 – otherwise the analyses neither support the claim nor do they reveal why detection as an additional objective actually helps. To establish that there is a center bias and trackers ignore the appearance term, the following two analyses could be done:

- Identify a set of test cases where capturing appearance is important (e.g. temporary occlusion) and demonstrate that your approach improves performance on these cases.

- For the table in Fig. 5, also show how a conventional tracker (i.e without object detection branch) performs on random targets. If the hypothesis is correct that the object detection objective during training reduces center bias and increases reliance on appearance, then a conventional tracker should show a smaller reduction due to random targets than your improved tracker does. The same quantitative analysis could also be done for existing state-of-the-art trackers, as it does not require training.

Minor Comments:
- Caption of Fig. 4: is instead of if?
- Figure 5
	- y-axis label: lower is better despite “robustness” suggesting the opposite
- Why do the numbers in the figure not match those in the table?
- What is the center bias baseline, i.e. what is the performance if center of search image is predicted without any network?


**Experience Assessment:**

I have read many papers in this area.

**Review Assessment: Checking Correctness Of Derivations And Theory:**

I assessed the sensibility of the derivations and theory.

**Review Assessment: Checking Correctness Of Experiments:**

I carefully checked the experiments.

**Review Assessment: Thoroughness In Paper Reading:**

I read the paper at least twice and used my best judgement in assessing the paper.

---

### Official Review · AnonReviewer3 · 2019-10-27
**Official Blind Review #3**

**Rating:** 1

**Review:**


= Summary
This paper proposes to learn a visual tracking network for an object detection loss as well as the ordinary tracking objective for enhancing the reliability of the tracking network. The main motivation is that, current state-of-the-art models based on the Siamese architecture often blindly predict the center of a search window as target location due to the bias in datasets and training strategies. This issue is alleviated in this paper by introducing an auxiliary task, target detection in the entire image space. The auxiliary task is conducted by another branch on top of the visual feature shared with the tracking branch. By learning to detect object in the entire image space, the shared feature extractor will be trained to capture discriminative and unique appearance features of target.


= Decision
Although the main motivation is convincing and the manuscript is well written, I would recommend to reject this submission mainly due to its limited contribution and weakness in experimental analysis.

(1) In the experiments, the practical benefit of adding the auxiliary detection task is demonstrated, but the final scores of the proposed model are clearly below those of current state of the art in terms of both reliability and accuracy. Further, it is not explained why the proposed model is worse than the other models in performance and what can be claimed as an advantage of the proposed method even in this situation. Also, I do not understand why the proposed model is not based on the current state of the art like SiamRPN++ but is built upon a manually designed/low-performance model.

(2) The experiments in Section 5 do not demonstrate the advantage of the proposed model at all. In Figure 6 and 7, the difference between the proposed model and its reduced version without the auxiliary task looks quite subtle, and it is hard to say which one is better than the others. In Figure 8, adding the auxiliary detection task results in even worse tracking performance.

(3) More qualitatively and quantitatively analysis should be done on the other tracking benchmarks and be compared with other tracking models recently proposed too.

**Experience Assessment:**

I have published one or two papers in this area.

**Review Assessment: Checking Correctness Of Derivations And Theory:**

I carefully checked the derivations and theory.

**Review Assessment: Checking Correctness Of Experiments:**

I carefully checked the experiments.

**Review Assessment: Thoroughness In Paper Reading:**

I read the paper thoroughly.

---

### Author Response · Authors · 2019-10-04
**Swapped legend in Figure 8.**

We noticed that the legend in Figure 8 is swapped. The blue curve should be labeled as "without detector" and orange curve should be labeled as "with detector". We will upload a revised version of the paper as soon as we are able to.

---

### Author Response · Authors · 2019-11-14
**Response to Reviewer #3**

Response to Reviewer #3
Thank you for your insightful review.
We would like to first clarify a point: Our claim is not that Siamese trackers “...blindly predict the center of a search window as target location…” but rather they have a tendency to predict the bounding box corresponding to the most salient object in the search window with some center bias. Indeed the strategy learned by the network of when to rely on the template and when to rely on saliency can be quite complicated as depicted by the experiment in Figure 7.

1 - The performance of a given tracker is not only dependent on the network output but also on post-processing. Namely the proposals generated by the network are selected according to hand-crafted heuristics. The representations learned by Siamese trackers are typically independent of the way the proposals are selected (certainly the case for SiamRPN++).The focus of this work is on analyzing the representations learned by the Siamese networks in these trackers. We will make this point clearer in the text. Our tracker results in only 7 more failures than SiamRPN++ (57 vs 50) on a total of 21356 frames. We believe this result to be sufficiently close to SoTA for our results to be considered relevant.

2 - Perhaps we should have made this clearer in the paper, but the purpose of Figures 6 & 7 is precisely to show that the detection task has little to no effect on reducing spatial bias. Where it does have an effect is on promoting more abstract learned target representations (e.g. temporally stale targets) as depicted in Figure 8. A small gain in average IoU can lead to a considerable gain in robustness (Figure 5). Please note that the legend was swapped in the original version of the paper, see our comment on OpenReview.

3 - We agree with the reviewer that additional analysis may provide further insights. Nevertheless, please note that the results illustrating the effect of the detector robustness (e.g. Figure 5) is already more statistically significant than what is typically reported in tracking literature. This is particularly important on datasets without standard train/validation splits such as VOT.  Specifically, we show robustness with error bars throughout training and not merely a final robustness value. What type of additional analysis would the reviewer suggest? In subsequent versions of the paper, we will also include additional perturbation analysis results for the open source version of the SiamRPN++ tracker in the final manuscript (early experimental results show SiamRPN++ has similar characteristics to our tracker - which is unsurprising given that the architectures of both trackers are both Siamese Correlation-based networks). We will also include a similar analysis on for a curated set of discrete events (e.g. saliency bias for frames labeled to contain occlusion).

---

### Author Response · Authors · 2019-11-14
**Response to Reviewer #1**

Response to Reviewer #1

Thank you for your insightful review. We address your concerns below:

"- Makes claims about Siamese trackers in general but only experiments with authors’ architecture”.

SiamRPN++ was open sourced only recently and therefore we could not run analysis directly on their architecture. We will do so for follow-up revision of the paper. Our architecture is similar to theirs, and initial experiments using SiamRPN++ suggests that the conclusions will transfer. The main difference between the two architectures is that we do not use multiple scales/aspect ratios anchors to generate our bounding box proposals as we found that it did not help performance. Indeed, we tried to reproduce their results exactly but could not get the stated results on OTB or VOT.

We agree with the reviewer that the hypothesis needs to be tested across other Siamese trackers as well.

"- Analyses in Figs. 6–8 not very helpful/conclusive”

Perhaps the results in Figs. 6-8 should be stated more clearly in the text. The aim of those figures was to show that the performance gained by adding the detector is attributed to learning more robust target representations. The aim of Figs. 6-7 is to show that the response to spatial perturbations between the models trained with and without the detector are virtually the same. Figure 8 shows a small, but stable improvement in the IoU when more abstract (temporally older) targets are fed into the network.

“- No test case where capturing appearance is important for tracking and authors’ approach helps”

Thanks for the helpful suggestion. We will attempt to perform such analysis for the final manuscript if accepted.

“- No comparison of tracker trained without detection objective on real vs. random targets”

We agree with the reviewer that such a model would be a useful baseline. A model trained end-to-end with random targets would learn to track as well as possible without making use of the target. Nevertheless, some of our experiments show that even models trained with standard target schemes learn to rely on the target in an “adaptive” fashion. Specifically, Figure 7 shows the IoU performance increases when the target is moved completely outside the receptive field of the network.

Thank you for the edit suggestions listed under “minor comments”. We will address these in the next version of the manuscript.

---

> ### Comment · AnonReviewer1 · 2019-11-14
> **The paper needs a revision**
>
> Thank you for your response. It seems like we mostly agree, but without seeing the additional analyses you promised I cannot change my assessment. So I would suggest revising the paper accordingly and re-submitting the revision in the future.

---

### Author Response · Authors · 2019-11-14
**Response to Reviewer #2**

Response to Reviewer #2

Thank you for the detailed feedback and careful review. Below we address your specific comments:

“However, the proposed solution of just integrating an additional detection task branch within the Siamese tracking architecture is naive. The main idea of integrating instance driven detection as an auxiliary task is borrowed from [1]”

Just to clarify, we do not claim that this is the first work to propose conditioning a full-frame detector on an exemplar image (as in [1], which we cite in our work). Our claim is that this is the first work to incorporate it in a tracking-specific architecture, trained end-to-end alongside a traditional center-crop Siamese tracking network. We would also like to highlight that the addition of the detector branch is simply to improve and regularize the latent representations used by the Siamese tracker. In this sense, the motivation for instance conditioned image detection is very different to those of [1] (which propose doing so to improve detector performance).

“Some recent works, such as [2, 3] have also investigated a similar problem of richer object representations for deep visual tracking. These approaches are desired to be discussed and empirically compared in order to fully validate the strength of the proposed approach.”


Thank you for the additional related work. We will cite these papers in the next version of the manuscript. While, discriminative (non Siamese cross-correlation based) trackers were outside the scope of the initial manuscript, it appears that inference code for DiMP has recently been released. As such, we will include perturbation analysis results for subsequent versions.


“Tracking datasets, such as VOT and OTB, provide additional analysis tools (i.e., attribute analysis) to thoroughly evaluate visual trackers. ... How does the proposed approach fare, compared to SOTA, on the subset of VOT image sequences that are labeled with occlusion?”

We thank the reviewer for this suggestion. We will include analysis correlating performance and characteristic to the included labels of VOT and OTB.

“The reviewer does not fully agree with this statement. A comprehensive empirical evaluation is crucial to fully access the merits of the contributions. State-of-the-art visual object trackers [2, 3, 4] achieve competitive tracking performance while being computationally efficient and fast. Therefore, a proper state-of-the-art comparison is desired to compare the proposed tracker with SOTA methods.”

As per our comment to Reviewer #3 and as we described in the paper, we choose specifically to not focus on SoTA results for VOT2018 and OTB but rather instead focus on analysis of Siamese trackers and the tracker performance when including full-frame image detection as an auxiliary loss. Note that the absolute number of failures between SiamRPN++ and this work is 7 (50 vs 57 failures out of 21356 frames). We believe this places our work sufficiently close to SoTA to make our experimental results relevant and interesting to the community.

“The reviewer recommends to perform additional experiments on other large-scale datasets, such as TrackingNet [5] and Lasot [6]”

We will include results on these datasets in the camera ready version of the paper if accepted. Note that analysis of tracker dynamics will be performed on the validation set of TrackingNet.

---

### Decision · Program_Chairs · 2019-12-19

**Decision:**

Reject

**Comment:**

This paper proposes to learn a visual tracking network for an object detection loss as well as the ordinary tracking objective for enhancing the reliability of the tracking network.  The reviewers were unanimous in their opinion that the paper should not be accepted to ICLR in its current form.  A main concern is that the proposed method shows improvement over a relatively weak base system.  Although the author response proposed to include additional analysis, but the reviewers felt that without the additional analysis already included it was not possible to change the overall review score.